# UAVDB: Point-Guided Masks for UAV Detection and Segmentation

## ABSTRACT

The widespread use of Unmanned Aerial Vehicles (UAVs) in surveillance, security, and airspace monitoring demands accurate and scalable detection methods. Progress, however, is limited by the lack of large-scale, high-resolution datasets with precise yet cost-efficient annotations. To address these challenges, we present UAVDB, a benchmark dataset for UAV detection and segmentation, built through a point-guided weak supervision pipeline. UAVDB leverages trajectory point annotations and RGB video frames from a multi-view drone tracking dataset captured by fixed cameras. We introduce Patch Intensity Convergence (PIC), a lightweight annotation method that converts trajectory points into high-fidelity bounding boxes, eliminating manual labeling while maintaining accurate spatial localization. From these boxes, we further derive instance segmentation masks using SAM2, enabling rich multi-task annotations with minimal supervision. UAVDB captures UAVs across diverse scales, ranging from clearly visible objects to nearly single-pixel instances, under challenging conditions. Additionally, PIC is lightweight and readily pluggable into other point-guided scenarios, making it easy to scale up dataset generation across various domains. We quantitatively show that PIC outperforms existing annotation techniques in IoU accuracy and efficiency. Finally, we benchmark several state-of-the-art (SOTA) YOLO detectors on UAVDB, establishing strong baselines for future research. UAVDB and all associated tools will be publicly released to accelerate point-guided detection and segmentation research.

## 1 INTRODUCTION

Precise UAV detection is critical for effective monitoring and threat response. While modern object detection algorithms, such as the YOLO-series (Jocher et al., 2023; Wang et al., 2024b;a; Jocher & Qiu, 2024; Tian et al., 2025; Lei et al., 2025), EfficientDet (Tan et al., 2020), and transformer-based detectors (Zhu et al., 2020b; Carion et al., 2020; Robinson et al., 2025), have shown remarkable progress in UAV-related tasks, their performance still heavily depends on the availability of high-quality annotations. Even state-of-the-art (SOTA) models tend to underperform when trained or evaluated on datasets with noisy labels or missing instances, particularly for tiny or fast-moving UAVs. Existing UAV-related datasets generally fall into two broad categories. The first focuses on ground-target detection, where aerial imagery is used to detect objects such as vehicles or pedestrians (Wang et al., 2021; Xu et al., 2022; Ding et al., 2021; Zhu et al., 2021; Razakarivony & Jurie, 2016; Kalra et al., 2019; Du et al., 2018; Hsieh et al., 2017; Robicquet et al., 2016; Mundhenk et al., 2016; Xia et al., 2018; Mandal et al., 2020; Barekatain et al., 2017; Li & Yeung, 2017; Zhu et al., 2020a; Bozcan & Kayacan, 2020; Wu et al., 2024). The second category comprises UAV-target datasets, where the UAV itself is the object of interest for detection or tracking. UAV-target datasets can be further divided into two subtypes: (1) UAV-to-UAV datasets, in which a camera mounted on one UAV tracks another in flight (Registry, 2023; Li et al., 2016; Rozantsev et al., 2015; Guo et al., 2025). These datasets require significant operational effort, as they involve flying multiple UAVs simultaneously and precisely locating target UAVs, making the data collection process time-consuming and skill-intensive. (2) Camera-to-UAV datasets, where the UAV is observed by an external camera that may be handheld, mobile, or fixed (but not on a UAV), including both RGB (Steininger et al., 2021; Pawełczyk & Wojtyra, 2020; Aksoy et al., 2019; Kashiyama et al.,

2020) and infrared (Dai et al., 2023; 2021; Dai et al., 2021; Huang et al., 2023; Jiang et al., 2021; Zhao et al., 2021; Zhu et al., 2023; Zhao et al., 2023) modalities.

While several RGB-based camera-to-UAV datasets have been introduced in recent years, they exhibit key limitations that hinder their applicability to real-world aerial surveillance, particularly for detecting small, distant UAVs in complex environments. These shortcomings underscore the need for a more representative and scalable benchmark, motivating the development of a new dataset. For instance, the dataset proposed in (Kashiyama et al., 2020) contains 600×600 resolution images annotated with three object categories: bird, helicopter, and airplane. However, it suffers from severe class imbalance, with only 74 bird instances compared to 1,392 helicopters and 190 airplanes. This imbalance leads to overfitting toward the dominant class, limiting generalization. Furthermore, while the images are sequentially ordered, they are extracted from extremely low-frame-rate videos, making the dataset unsuitable for temporal modeling or video-based tracking. The dataset presented in (Pawełczyk & Wojtyra, 2020) includes videos with original resolutions ranging from 640×480 to 4K. However, all training and testing images are downscaled to 640×480, constraining the detection of tiny UAVs where high-resolution input is essential. Another dataset (Steininger et al., 2021) spans a wide range of image resolutions from 192×144 to 3840×2160, yet many images are now inaccessible, undermining reproducibility and long-term benchmarking. Other efforts, such as (Özel, 2018) and (Aksoy et al., 2019), provide 1,359 and approximately 4,000 images with resolutions of 1280×720 and between 300×168 and 4633×3089, respectively. However, both lack temporal coherence, as their images are not sourced from continuous video streams, limiting their suitability for motion-based tasks such as trajectory estimation and temporal modeling. Several additional datasets (aydin, 2024; WorkspaceTest1, 2025; flippinggreatwodgesofdroneimages1, 2022; ConcordiaNAVLab, 2023; SegmentDrones, 2023; Drone, 2024; Gourish, 2022; Jog, 2023) target UAV-related vision tasks but still fall short for long-range surveillance and temporally-aware applications. Most of the aforementioned datasets lack high-resolution temporal data, diverse environmental conditions, and consistent annotation quality. Moreover, they predominantly feature large UAVs captured from ground-level or short-range viewpoints, settings that differ significantly from real-world surveillance scenarios where UAVs typically appear small, distant, and often partially occluded within cluttered aerial scenes.

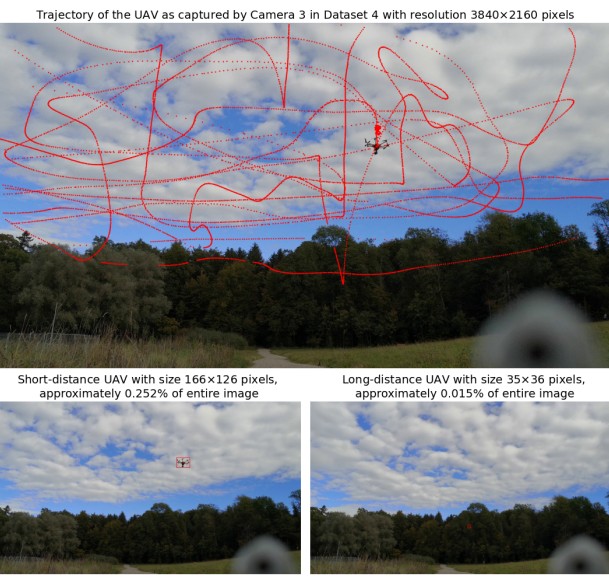

Figure 1: UAV trajectory captured by Camera 3 in Dataset 4 at 3840×2160 resolution in (Li et al., 2020). The yellow path represents the UAV's trajectory. On the left, the UAV appears at a short distance with a size of 166×126 pixels, occupying approximately 0.252% of the total image area. On the right, the UAV is shown at a long distance, with a size of 35×36 pixels, covering approximately 0.015% of the entire image. This figure shows the varying visibility of the UAV depending on its distance from the camera.

Table 1: Summary of dataset characteristics in (Li et al., 2020). The table displays the number of frames and resolution for each camera across different datasets. Each cell lists the number of frames followed by the resolution in pixels.

| Camera \ Dataset | 1 | 2 | 3 | 4 | 5 |
|---|---|---|---|---|---|
| 0 | 5334 / 1920×1080 | 4377 / 1920×1080 | 33875 / 1920×1080 | 31075 / 1920×1080 | 20970 / 1920×1080 |
| 1 | 4941 / 1920×1080 | 4749 / 1920×1080 | 19960 / 1920×1080 | 15409 / 1920×1080 | 28047 / 1920×1080 |
| 2 | 8016 / 1920×1080 | 8688 / 1920×1080 | 17166 / 3840×2160 | 15678 / 1920×1080 | 31860 / 2704×2028 |
| 3 | 4080 / 1920×1080 | 4332 / 1920×1080 | 14196 / 1440×1080 | 10933 / 3840×2160 | 31992 / 1920×1080 |
| 4 | – | – | 18900 / 1920×1080 | 17640 / 1920×1080 | 21523 / 2288×1080 |
| 5 | – | – | 28080 / 1920×1080 | 32016 / 1920×1080 | 17550 / 1920×1080 |
| 6 | – | – | – | 11292 / 1440×1080 | – |

To overcome the limitations of existing RGB-based camera-to-UAV datasets, we introduce UAVDB, a high-resolution dataset of multiscale UAVs captured under diverse and challenging conditions using static ground-based cameras. Designed for long-range aerial surveillance, UAVDB emphasizes small and distant targets in realistic scenarios such as monitoring restricted zones or critical infrastructure, providing a strong benchmark for detection and tracking under real-world constraints. UAVDB is built upon the multi-view drone tracking dataset (Li et al., 2020), which was developed for 3D trajectory reconstruction using unsynchronized consumer cameras with unknown viewpoints. This dataset offers high-resolution RGB videos with corresponding 2D UAV locations, forming a solid foundation for addressing gaps in prior UAV datasets. We propose Patch Intensity Convergence (PIC) to generate object detection annotations, a technique that automatically derives accurate 2D bounding boxes from trajectory points. We then leverage the Segment Anything Model v2 (SAM2) (Ravi et al., 2024), using the PIC-generated boxes as prompts to produce instance masks. Notably, this annotation pipeline requires no manual labeling, from trajectory points to masks. Furthermore, we intentionally avoid using point-based prompts directly with SAM2, as the 2D trajectory points are not always spatially precise, often leading to degraded segmentation quality. This limitation and its implications are discussed in detail in subsequent sections. To illustrate the diversity of UAV scales in the dataset, we visualize representative UAV trajectories alongside human-labeled bounding boxes across different size ranges, as shown in Fig. 1. A summary of the dataset characteristics in the multi-view drone tracking dataset (Li et al., 2020) is provided in Tab. 1, including the number of frames and camera resolutions across different sequences. In this paper, our contributions are as follows:

1. We introduce UAVDB, a high-resolution RGB video dataset for UAV detection and segmentation, featuring multiscale targets in complex and dynamic environments. UAVDB is constructed by first transforming trajectory data (Li et al., 2020) into precise bounding box annotations using the proposed Patch Intensity Convergence (PIC) method, followed by applying SAM2 (Ravi et al., 2024) to generate high-quality masks across video frames.

2. We validate the efficiency of PIC through experiments measuring IoU accuracy and runtime performance. Additionally, we provide a comprehensive benchmark of UAVDB using SOTA YOLO-series detectors, including YOLOv8 (Jocher et al., 2023), YOLOv9 (Wang et al., 2024b), YOLOv10 (Wang et al., 2024a), YOLOv11 (Jocher & Qiu, 2024), YOLOv12 (Tian et al., 2025), and YOLOv13 (Lei et al., 2025).

## 2 RELATED WORK

### 2.1 POINT-GUIDED WEAK SUPERVISION

Recent research has demonstrated the effectiveness of point-level annotations as a weak form of supervision across various computer vision tasks. In object detection and oriented object detection, numerous works have explored using single-point supervision to replace or augment bounding box annotations (Zhang et al., 2023; Tan & Wu, 2024b;a; Wang et al., 2023; Yu et al., 2024; Luo et al., 2024; Zhang et al., 2022; Ge et al., 2023; Chen et al., 2021; Tufekci Dogan et al., 2024; Cui et al., 2025; Ying et al., 2023; Li et al., 2023; 2024; May et al., 2024; Liu et al., 2023; Wong, 2024; Aggrawal et al., 2023). These methods reduce annotation cost, including in remote sensing and infrared imaging, but often depend on complex training pipelines involving point-to-box

regressors, orientation estimation modules, or synthetic priors. In the segmentation domain, point annotations have been used to supervise instance masks (Chen et al., 2025; Kim et al., 2023), refine object boundaries (Breznik et al., 2024), or generate dense proposals (Yao et al., 2024); however, segmentation quality often degrades on small or irregularly shaped objects without additional supervision. In 3D object detection, recent methods incorporate spatial point priors to bridge 2D imagery and 3D reasoning (Gao et al., 2024), but typically require multimodal data fusion and heavy model customization. Despite the promise of these approaches, most require end-to-end model training, suffer from generalization issues across domains, or are computationally intensive. In contrast, our work proposes a training-free, plug-and-play pipeline that operates directly on trajectory points and raw video frames, offering robust and scalable annotation generation without model retraining or domain-specific tuning.

## 2.2 BOUNDING BOX EXTRACTION VIA SEGMENTATION

Generating high-quality bounding box annotations for UAVs of varying sizes in video data using only trajectory information is a critical first step, as illustrated in Fig. 1. While learning-based methods may yield accurate results, they require substantial design and training effort. We focus on simpler, out-of-the-box techniques for bounding box extraction to reduce complexity. A naive solution is to assign fixed-size boxes centered at trajectory points; however, this lacks adaptability to UAV scale variations. A natural extension is to segment the region around each point and extract a bounding box from the resulting mask. Traditional image thresholding (Al-Amri et al., 2010) is a commonly used method for this task, but it struggles in low-contrast scenes and often requires manual parameter tuning. GrabCut (Rother et al., 2004) improves upon this by iteratively refining the foreground mask, though it remains computationally expensive and inefficient for large-scale annotation. Deep learning-based variants such as DeepGrabCut (Xu et al., 2017) further increase computational costs. More recent methods like SAM (Kirillov et al., 2023) and SAM2 (Ravi et al., 2024) enable zero-shot segmentation using point prompts. However, their effectiveness degrades in UAV-specific domains due to domain shifts and the spatial imprecision of trajectory points, often resulting in inaccurate or unstable segmentations. These limitations are illustrated in the top portion of Fig. 2, which compares the bounding boxes generated by various methods with human-labeled annotations across different datasets and camera viewpoints.

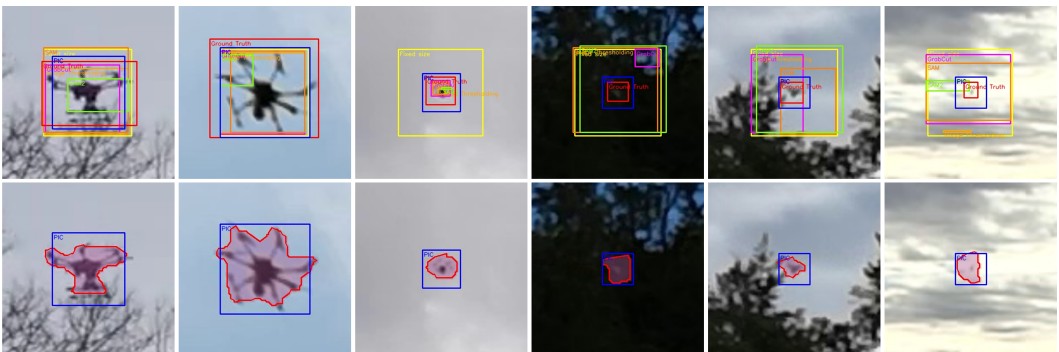

Figure 2: Top: Comparison of bounding box outputs from multiple methods, including fixed-size, image thresholding (Al-Amri et al., 2010), GrabCut (Rother et al., 2004), SAM (Kirillov et al., 2023), SAM2 (Ravi et al., 2024), and the proposed PIC (blue), shown alongside human-labeled ground truth annotations (red). Bottom: Segmentation masks generated by SAM2 (Ravi et al., 2024) using the PIC-derived bounding box as a prompt.

## 3 METHODOLOGY

To construct UAVDB with minimal manual effort, we propose an automated annotation pipeline that transforms 2D trajectory points into high-quality mask labels. It comprises two components: (1) bounding box generation via Patch Intensity Convergence (PIC), and (2) mask generation using Segment Anything Model v2 (SAM2) (Ravi et al., 2024).

### 3.1 Bounding Box Generation via PIC

The PIC technique extracts UAV bounding boxes from trajectory annotations via an adaptive inward-outward expansion, ensuring efficient localization without relying on external models or predefined dimensions. The process consists of four steps: initialization, iterative expansion, patch intensity calculation, and convergence assessment.

#### 3.1.1 Initialization

Given a trajectory point $(x_0, y_0)$, the bounding box is initialized as a square region $B_0$ of size $w_0 \times h_0$:

$$B_0 = \{(x,y) \mid x_0 - w_0/2 \leq x \leq x_0 + w_0/2, \ y_0 - h_0/2 \leq y \leq y_0 + h_0/2\}.$$

#### 3.1.2 Iterative Expansion

At each step $t$, the bounding box expands outward by a fixed size $\delta$ in all directions:

$$w_{t+1} = w_t + \delta, \quad h_{t+1} = h_t + \delta, \quad t = 0, 1, \ldots$$

The expanded region $B_{t+1}$ captures a progressively larger area around the trajectory point.

#### 3.1.3 Patch Intensity Calculation

The mean pixel intensity at each step inside the bounding box is computed as:

$$\mu_t = \frac{1}{|B_t|} \sum_{(x,y) \in B_t} I(x,y).$$

where $I(x,y)$ denotes the pixel intensity at $(x,y)$.

#### 3.1.4 Convergence Assessment

Expansion halts when the intensity change between consecutive iterations falls below a threshold $\epsilon$:

$$|\mu_{t+1} - \mu_t| < \epsilon.$$

This criterion ensures that further expansion does not significantly contribute to capturing UAV-relevant pixels, marking the final bounding box boundary.

We apply the PIC technique to the videos and trajectory data from (Li et al., 2020), using an initial patch size of $w_0 = h_0 = 8$ pixels, an expansion step of $\delta = 5$ pixels, and a convergence threshold of $\epsilon = 4$. As shown in Fig. 3, the middle column visualizes the stepwise expansion and corresponding pixel intensity values across different datasets, illustrating PIC's robustness in challenging conditions. The rightmost column provides reference images indicating UAV size as a percentage of the total image area. PIC successfully localizes UAVs across a wide range of scales, from large instances ($53 \times 52$ pixels around 0.133% of the image) to tiny ones ($13 \times 13$ pixels around 0.008% of the image), resulting in high-fidelity bounding box annotations. For UAVDB, we sample one frame every ten frames (around 10% of the footage) from the sequences listed in Tab. 1. This results in a dataset comprising 10,763 training images, 2,720 validation images, and 4,578 test images, as summarized in Tab. 2. Dataset 5 from (Li et al., 2020), which lacks 2D trajectory data, is treated as an unseen scenario, with segmentation predictions demonstrated in the experimental section. Notably, our framework supports flexible adjustment of the frame extraction rate, enabling users to scale the dataset size according to application needs.

### 3.2 Mask Generation using SAM2

To extend UAVDB with segmentation annotations, we leverage SAM2 (Ravi et al., 2024), a powerful zero-shot segmentation model capable of generating instance masks given a bounding box or point prompt, inspired in part by (Mukherjee et al., 2025). Our approach uses bounding boxes generated by PIC as box prompts to guide SAM2, enabling automated and consistent mask extraction across diverse scenes. This box-based prompting is essential. While SAM2 supports point prompts, we observe that trajectory points are often spatially imprecise due to motion blur, occlusion, or annotation

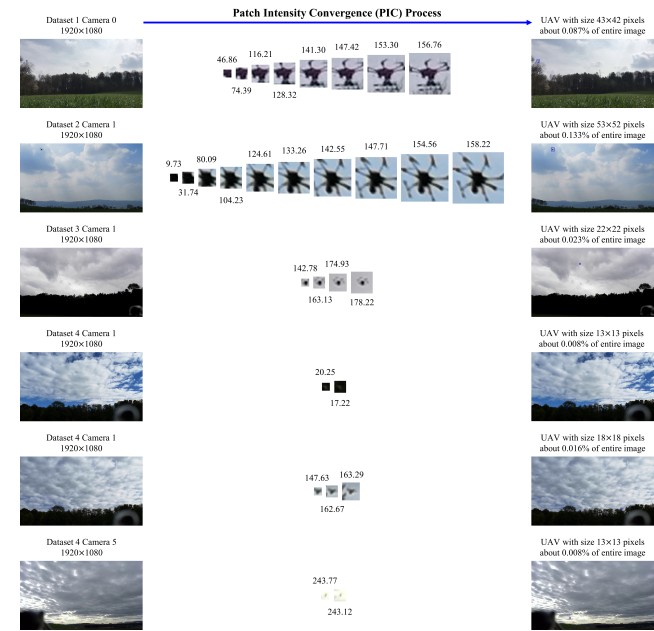

Figure 3: Stepwise illustration of the PIC process across datasets and camera views. The middle column shows iterative bounding box expansion with corresponding intensity values. The rightmost column presents the final PIC annotations, including UAV size and aspect ratio for each scenario.

Table 2: Overview of the UAVDB constructed using the proposed PIC approach. The table shows the distribution of images across different datasets and camera configurations, specifying the number of images used for training, validation, and testing.

| Camera \ Dataset | 1 | 2 | 3 | 4 |
|---|---|---|---|---|
| 0 | train / 291 | test / 237 | train / 3190 | test / 2355 |
| 1 | valid / 303 | train / 343 | train / 841 | train / 416 |
| 2 | train / 394 | train / 809 | valid / 1067 | train / 701 |
| 3 | test / 348 | valid / 426 | train / 638 | train / 727 |
| 4 | – | – | test / 1253 | valid / 924 |
| 5 | – | – | train / 1303 | train / 1110 |
| 6 | – | – | – | test / 385 |

noise. Directly applying point prompts frequently leads to poor or off-target masks, particularly for small UAVs, as shown in the upper row of Fig. 2. In contrast, PIC-derived boxes provide spatially localized, high-confidence regions that allow SAM2 to focus on a constrained area, resulting in more accurate segmentation masks. These mask annotations complement the detection labels, making UAVDB suitable for object and instance segmentation tasks. As shown in the bottom row of Fig. 2, the SAM2-generated masks often better capture object shape than PIC bounding boxes, especially for larger UAVs. However, as shown in the rightmost subplot in the bottom row, the masks may not tightly align with object boundaries for extremely small UAVs, yet they perform comparably to bounding boxes. This highlights the strengths and limitations of mask-based annotations for tiny object segmentation.

## 4 EXPERIMENTAL RESULTS

We first evaluate the effectiveness of the proposed PIC approach in terms of Intersection over Union (IoU) and runtime efficiency, compared to other annotation methods. We then present comprehensive benchmark results on UAVDB using YOLO-series detectors.

Table 3: Comparison of different UAV bounding box extraction methods regarding average IoU and runtime (seconds).

| Methods | Average IoU ↑ | Runtime (s) ↓ |
|---|---|---|
| human-labeled | 1.000 | 19.00 |
| Fixed-size | 0.278 | 0.007 |
| Thresholding (Al-Amri et al., 2010) | 0.316 | 0.009 |
| GrabCut (Rother et al., 2004) | 0.425 | 2.423 |
| SAM (Kirillov et al., 2023) | 0.249 | 0.484 |
| SAM2 (Ravi et al., 2024) | 0.119 | 0.229 |
| **PIC (ours)** | **0.464** | **0.007** |

## 4.1 ANNOTATION ACCURACY AND RUNTIME EFFICIENCY

Firstly, human-labeled bounding boxes serve as the ground truth annotations. For the fixed-size and thresholding (Al-Amri et al., 2010) baselines, we use a 50×50 region and set the threshold to 150, based on empirical tuning for best performance. GrabCut (Rother et al., 2004), SAM (Kirillov et al., 2023), and SAM2 (Ravi et al., 2024) are implemented using OpenCV, ViT-B SAM, and Hiera-L SAM2 pre-trained models, respectively. As shown in Tab. 3, the proposed PIC method achieves the highest IoU while maintaining a minimal runtime of just 0.007 seconds, comparable to the fixed-size approach, and is approximately 2700× faster than manual annotation. This confirms that PIC introduces negligible computational overhead relative to the time required for image I/O. In contrast, manual annotation takes an average of 19 seconds per bounding box, making it impractical for large-scale datasets with tiny objects. Despite the SAM series' advanced segmentation capabilities, SAM and SAM2 perform poorly when directly using point prompts, yielding the lowest IoU scores due to domain shifts and imprecise prompt localization. These results highlight the effectiveness of PIC in delivering accurate and efficient bounding box annotations, making it well-suited for large-scale and even real-time UAV applications.

## 4.2 BENCHMARK ON UAVDB

We benchmark the proposed UAVDB using YOLO-series detectors, including YOLOv8 (Jocher et al., 2023), YOLOv9 (Wang et al., 2024b), YOLOv10 (Wang et al., 2024a), YOLOv11 (Jocher & Qiu, 2024), YOLOv12 (Tian et al., 2025), and YOLOv13 (Lei et al., 2025). All experiments were conducted on a high-performance computing (HPC) system (Meade et al., 2017) equipped with an NVIDIA A100 GPU (80 GB memory). Models were trained using an input size of 640, a batch size of 32, for 100 epochs, with eight dataloader workers. Mosaic augmentation was applied during training, excluding the final 10 epochs. Each model was fine-tuned using its official pre-trained weights. As shown in Tab. 4, we summarize training time, inference speed, model size (parameters and FLOPs), and average precision (AP) on both validation and test sets.

In addition to object detection, we trained the YOLOv12n-seg (Tian et al., 2025) model for instance segmentation with an image size of 1920, a batch size of 12, and 100 training epochs. The large image size facilitates better mask detail learning. Training took approximately one and a half days, and during inference, the model processes images at an average speed of 9.0 milliseconds per frame. The model contains 2.761M parameters and requires 9.7 GFLOPs per forward pass. Both bounding box and mask precision results are presented in Tab. 5, where the performance gap between the validation and test sets suggests potential overfitting. This issue can be mitigated by increasing the dataset size, a straightforward process enabled by UAVDB's flexible frame extraction rate.

We further visualize the generalization capability of the trained YOLOv12n-seg model on Dataset 5, which was entirely excluded from training and validation. Unlike typical unseen splits with similar data distributions, Dataset 5 represents a distinct scenario, making detection and segmentation more challenging. As shown in Fig. 4, we present sequential predictions from Camera 3 (top row) and Camera 5 (bottom row) across consecutive frames. Despite the UAVs being small, blurry, and often embedded in complex backgrounds, the model demonstrates strong generalization, with well-aligned bounding boxes and segmentation masks that tightly fit the UAVs. Leveraging the video-

Table 4: Performance comparison of YOLOv8 (Jocher et al., 2023), YOLOv9 (Wang et al., 2024b), YOLOv10 (Wang et al., 2024a), YOLOv11 (Jocher & Qiu, 2024), YOLOv12 (Tian et al., 2025), and YOLOv13 (Lei et al., 2025) models trained on UAVDB using PIC-generated bounding boxes for the object detection task.

| Model | Training Time (hours:mins:sec) | Inference Time (per image, ms) | #Param. (M) | FLOPs (G) | $AP_{50}^{val}$ | $AP_{50-95}^{val}$ | $AP_{50}^{test}$ | $AP_{50-95}^{test}$ |
|---|---|---|---|---|---|---|---|---|
| YOLOv8n | 01:40:31 | 0.9 | 2.685 | 6.8 | 0.829 | 0.522 | 0.789 | 0.450 |
| YOLOv8s | 01:55:05 | 1.2 | 9.828 | 23.3 | 0.814 | 0.545 | 0.796 | 0.450 |
| YOLOv8m | 02:43:08 | 1.8 | 23.203 | 67.4 | 0.809 | 0.538 | 0.827 | 0.526 |
| YOLOv8l | 03:54:44 | 2.6 | 39.434 | 145.2 | 0.830 | 0.563 | 0.836 | 0.544 |
| YOLOv8x | 04:33:08 | 3.5 | 61.597 | 226.7 | 0.820 | 0.554 | 0.728 | 0.448 |
| YOLOv9t | 02:53:11 | 2.5 | 2.617 | 10.7 | 0.839 | 0.501 | 0.848 | 0.508 |
| YOLOv9s | 03:05:02 | 2.6 | 9.598 | 38.7 | 0.819 | 0.517 | 0.834 | 0.484 |
| YOLOv9m | 05:08:28 | 4.1 | 32.553 | 130.7 | 0.840 | 0.507 | 0.858 | 0.522 |
| YOLOv9c | 06:17:08 | 5.3 | 50.698 | 236.6 | 0.851 | 0.544 | 0.851 | 0.504 |
| YOLOv9e | 08:00:05 | 6.6 | 68.548 | 240.7 | 0.755 | 0.414 | 0.768 | 0.383 |
| YOLOv10n | 02:05:39 | 0.7 | 2.695 | 8.2 | 0.764 | 0.492 | 0.731 | 0.417 |
| YOLOv10s | 02:23:03 | 1.2 | 8.036 | 24.4 | 0.817 | 0.530 | 0.823 | 0.516 |
| YOLOv10m | 03:06:59 | 1.8 | 16.452 | 63.4 | 0.798 | 0.531 | 0.821 | 0.536 |
| YOLOv10b | 03:29:18 | 2.1 | 20.413 | 97.9 | 0.801 | 0.517 | 0.760 | 0.467 |
| YOLOv10l | 04:04:22 | 2.5 | 25.718 | 126.3 | 0.774 | 0.502 | 0.842 | 0.517 |
| YOLOv10x | 05:14:07 | 3.5 | 31.586 | 169.8 | 0.771 | 0.507 | 0.693 | 0.431 |
| YOLOv11n | 01:50:00 | 0.9 | 2.582 | 6.3 | 0.847 | 0.527 | 0.856 | 0.539 |
| YOLOv11s | 02:07:01 | 1.2 | 9.413 | 21.3 | 0.826 | 0.553 | 0.885 | 0.578 |
| YOLOv11m | 03:07:40 | 1.9 | 20.031 | 67.6 | 0.827 | **0.588** | 0.843 | 0.578 |
| YOLOv11l | 04:09:45 | 2.4 | 25.280 | 86.6 | 0.810 | 0.555 | 0.798 | 0.517 |
| YOLOv11x | 05:20:38 | 3.6 | 56.828 | 194.4 | 0.812 | 0.560 | 0.782 | 0.534 |
| YOLOv12n | 02:15:38 | 1.8 | 2.557 | 6.3 | 0.857 | 0.544 | 0.848 | 0.531 |
| YOLOv12s | 02:44:29 | 2.0 | 9.231 | 21.2 | 0.869 | 0.566 | 0.882 | 0.565 |
| YOLOv12m | 03:34:36 | 2.6 | 20.106 | 67.1 | 0.866 | 0.567 | 0.886 | 0.584 |
| YOLOv12l | 05:10:15 | 3.1 | 26.340 | 88.5 | 0.870 | 0.584 | 0.875 | **0.590** |
| YOLOv12x | 06:35:47 | 3.9 | 59.045 | 198.5 | **0.879** | 0.576 | **0.896** | 0.569 |
| YOLOv13n | 03:23:00 | 1.6 | 2.448 | 6.2 | 0.833 | 0.541 | 0.795 | 0.505 |
| YOLOv13s | 04:15:04 | 2.1 | 9.530 | 21.3 | 0.852 | 0.555 | 0.804 | 0.496 |
| YOLOv13l | 10:07:28 | 5.5 | 27.514 | 88.1 | 0.860 | 0.554 | 0.826 | 0.540 |
| YOLOv13x | 13:40:58 | 8.3 | 63.886 | 198.7 | 0.846 | 0.568 | 0.836 | 0.556 |

Table 5: Performance of YOLOv12n-seg (Tian et al., 2025) trained on UAVDB with SAM2-generated masks for instance segmentation.

| Model | Box | | | | Mask | | | |
|---|---|---|---|---|---|---|---|---|
| | $AP_{50}^{val}$ | $AP_{50-95}^{val}$ | $AP_{50}^{test}$ | $AP_{50-95}^{test}$ | $AP_{50}^{val}$ | $AP_{50-95}^{val}$ | $AP_{50}^{test}$ | $AP_{50-95}^{test}$ |
| YOLOv12n-seg | 0.946 | 0.608 | 0.936 | 0.519 | 0.941 | 0.523 | 0.756 | 0.307 |

based nature of UAVDB, we move beyond static detection to continuous tracking, enabling richer and more realistic evaluation than traditional image-level detection.

## 4.3 ANALYSIS AND DISCUSSION

Here, we analyze the sensitivity of PIC to its three parameters: initial patch size $(w_0, h_0)$, expansion step $\delta$, and convergence threshold $\epsilon$. Larger initial patches accelerate convergence but may include background clutter, while smaller patches better localize tiny UAVs at the cost of more iterations. Similarly, larger step sizes $\delta$ reduce computation but risk overshooting object boundaries, whereas smaller values yield tighter fits at a higher cost. The convergence threshold $\epsilon$ controls the stopping condition: stricter thresholds improve bounding-box fidelity but provide diminishing returns in average precision. These trade-offs suggest that PIC can be tuned efficiently by aligning parameter scales with object sizes and motion patterns across datasets.

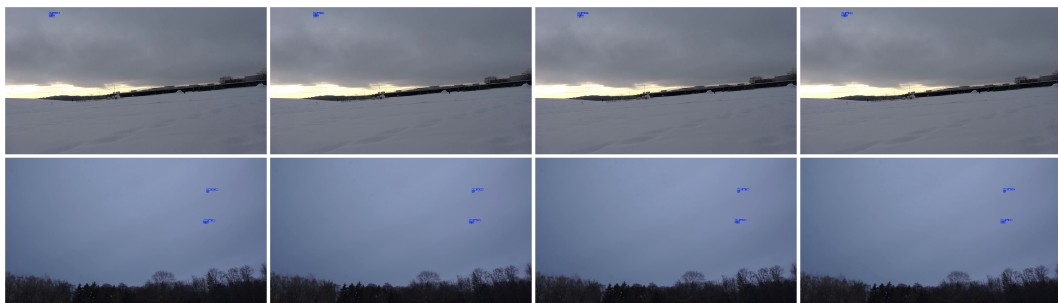

Figure 4: Sequential tracking results predicted by YOLOv12n-seg (Tian et al., 2025) on the entirely unseen Dataset 5. Top: Camera 3. Bottom: Camera 5. Left to right shows consecutive video frames.

Another limitation of PIC is that it produces near-square bounding boxes, which may misalign with elongated UAVs or objects in other domains. However, this can be effectively addressed by the SAM2 refinement stage. As shown in Tab. 4 versus Tab. 5, detectors trained on refined annotations achieve consistently higher accuracy. For example, YOLOv12n trained on PIC boxes attains $\text{AP}_{50}^{val} = 0.857$, $\text{AP}_{50-95}^{val} = 0.544$, $\text{AP}_{50}^{test} = 0.848$, and $\text{AP}_{50-95}^{test} = 0.531$, whereas YOLOv12n-seg trained on PIC with SAM2 annotations improves to $0.946$, $0.608$, $0.936$, and $0.519$, respectively. These gains demonstrate that SAM2 refinement corrects systematic misalignments by converting coarse square boxes into more faithful rectangular ones. The complete pipeline, therefore, not only mitigates the square-box limitation but also enhances bounding-box annotation quality.

In summary, this analysis shows that (i) PIC parameters can be systematically tuned to match dataset resolution and object scale, and (ii) SAM2 refinement effectively compensates for shape mismatches, yielding measurable improvements in both detection and segmentation benchmarks.

## 5 CONCLUSION

We introduced UAVDB, a high-resolution, video-based benchmark explicitly designed for RGB-based camera-to-UAV monitoring in long-range aerospace surveillance scenarios. UAVDB addresses critical gaps in existing datasets, which often lack the resolution, diversity, and temporal continuity necessary to detect and track small, distant UAVs in complex environments. Built upon a lightweight and scalable point-guided weak supervision pipeline, UAVDB eliminates manual labeling once trajectory points are available. Our proposed Patch Intensity Convergence (PIC) method accurately derives bounding boxes from these points, which are then used to prompt SAM2 for generating high-quality instance masks, enabling fully automated annotation with minimal human effort. Beyond detection and segmentation, UAVDB's video-based nature supports flexible scaling via adjustable frame sampling and enables temporal tasks such as tracking, making it significantly more versatile than conventional static image benchmarks. Furthermore, the modular PIC with the SAM2 pipeline is transferable and can be integrated into other point-guided vision tasks beyond UAV surveillance. In conclusion, UAVDB offers a valuable foundation for developing and benchmarking robust detection, segmentation, and tracking methods under realistic conditions, and expects the annotation pipeline to advance research in weakly supervised, domain-adaptive, and video-aware computer vision.

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
