# OpenReview forum: "UAVDB: Point-Guided Masks for UAV Detection and Segmentation"
_ICLR.cc/2026/Conference — ICLR 2026 Conference Withdrawn Submission_

### Official Review · Reviewer_zpq3 · 2025-10-27

**Soundness:** 2
**Presentation:** 2
**Contribution:** 1
**Rating:** 2
**Confidence:** 3

**Summary:**

This paper describes a new layer of annotations on top the the UAV tacking dataset of Li et al. (2020). These annotations are created from point annotations stemming from the 2020 dataset by using a point-based object detector (the proposed Point Intensity Convergence, PIC) and SAM2 to turn the bounding boxes into segmentation masks.
The proposed PIC is a heuristic for finding a good bounding box size assuming that: (1) the central point of the BB is given, (2) the BB is square and (3) the object is darker than the surroundings. It consists of starting with some fixed initial size and increasing the size a fix step at a time. The average pixel intensity is measured and an elbow in the intensity is detected when the (positive) change in intensity between one step and the next becomes smaller than a fixed threshold.

**Strengths:**

- The proposed approach is simple and easy to grasp.
- The paper is also easy to read.
- The proposed PIC is computationally very efficient.

**Weaknesses:**

1. The proposed dataset builds heavily on the dataset of Li at al. (2020), making the contribution from the dataset perspective very limited.
2. The main contribution is thus PIC. The proposed method is a simple heuristic that is very efficient to compute, but that is hardly a general method and is not sufficiently evaluated. More specifically:
- PIC is compared against alternative method on a human-annotated dataset. There are insufficient details about this dataset to assess it’s qualities. This human-annotated benchmark should be one of the central pillars of the paper, since it is the only evaluation of the main contribution of the paper.
- PIC’s simplicity also means it will fail to generalize when the main (unwritten) assumptions don’t hold, such as when the object is brighter than the background (thermal imagery? UAV’s with lights? Night scene?) or the desired BB is not square.
- There are, at least, three important hyperparameters: init size, step size and threshold, but I couldn’t find any sensitivity analysis in the results.
3. In the introduction and related works there are some general statements about the limitations of existing datasets (lines 74 to 76) and methods (lines 168 to 170), without pointing out specific limitations of each of the cited works. A table clearly showing the pros and cons of each work would be very useful here.
4. Overall, I doubt this work fulfills the publication standards detailed in the ICLR reviewer guidelines: there is no novel idea being proposed, and the paper would be of interest to a very small fraction of the ICLR readership. Given the limited contribution, both in terms of datasets and methods, I would suggest to find a suitable workshop to submit this work to.

**Questions:**

I have no outstanding question for the authors. However, I would strongly suggest them to address weaknesses 2 and 3 (hyperparameter sensitivity study and better positioning wrt sota) before submitting to another venue, be it a workshop at an ML conference or other.

---

### Official Review · Reviewer_Z4yc · 2025-10-30

**Soundness:** 3
**Presentation:** 3
**Contribution:** 2
**Rating:** 4
**Confidence:** 4

**Summary:**

This paper introduces UAVDB, a high-resolution dataset for UAV detection and segmentation built upon an existing multi-view drone-tracking video collection. It proposes a lightweight automatic annotation method called Patch Intensity Convergence (PIC), which generates bounding boxes from trajectory points and uses SAM2 to produce instance masks. The authors benchmark several YOLO-series detectors to evaluate detection and segmentation performance under this weakly supervised labeling pipeline. Overall, the work focuses on automated annotation and benchmark construction for UAV perception tasks.

**Strengths:**

1.The paper is clearly written, with logical structure and high-quality figures that effectively illustrate the proposed pipeline.

2.The proposed automatic annotation pipeline (PIC + SAM2) is simple, fast, and reproducible, offering practical value for reducing manual labeling cost in UAV datasets.

3.The work provides a unified baseline for UAV detection under weak supervision and can serve as a useful engineering reference for future data construction research.

**Weaknesses:**

1.Limited novelty: The core algorithm, Patch Intensity Convergence (PIC), is a simple heuristic based on iterative patch expansion and mean intensity convergence. It lacks theoretical justification or methodological innovation and is conceptually similar to classic region-growing and thresholding techniques.

2.Dataset originality is low: UAVDB is mainly derived from an existing dataset (Li et al., 2020) with automatically generated annotations rather than new data collection. Thus, the dataset’s contribution lies more in re-labeling than in creating new sensing modalities or tasks.

3.Dependence on external models: The use of SAM2 for mask generation offloads much of the contribution to a pretrained foundation model, reducing the authors’ own technical novelty.

4.Limited experimental depth: Although many YOLO versions are benchmarked, the evaluation focuses only on detection accuracy and speed. There is no cross-dataset validation, robustness analysis, or ablation to show how PIC behaves under different lighting or motion conditions.

5.Lack of theoretical or practical insight: The paper does not explain why the proposed intensity-based convergence is superior conceptually, nor how it might generalize to other weak-supervision or multimodal UAV scenarios.

**Questions:**

1.Could the authors further explain the theoretical motivation behind Patch Intensity Convergence? Why does the intensity convergence threshold effectively delineate UAV boundaries? Have alternative local statistics (e.g., variance, gradient, or texture features) been explored instead of mean intensity?

2.How is PIC fundamentally different from classic region-growing or adaptive thresholding techniques? Could the authors provide mathematical analysis or comparative experiments to better highlight its uniqueness?

3.Since UAVDB is largely derived from Li et al. (2020), how do the authors plan to strengthen its independent contribution? Are there plans to incorporate multimodal data (e.g., infrared, depth) or temporal annotations in future versions?

4.Given that the pipeline heavily relies on SAM2 for mask generation, how would the overall performance change if SAM2 were replaced by a smaller or non-foundation model?

5.Have the authors tested this automatic annotation pipeline in non-UAV domains (e.g., traffic surveillance or small-object detection)? Demonstrating cross-domain generalization would significantly increase the paper’s impact.

---

### Official Review · Reviewer_nyzK · 2025-11-01

**Soundness:** 2
**Presentation:** 3
**Contribution:** 3
**Rating:** 2
**Confidence:** 4

**Summary:**

This paper introduces UAVDB, a new benchmark dataset for UAV detection and segmentation, designed to address the limitations of existing datasets for long-range aerial surveillance. This paper proposes Patch Intensity Convergence (PIC) to generate high-quality bounding boxes from trajectory points, which are then used to prompt SAM2 for creating instance segmentation masks. The utility of UAVDB and the PIC method is validated through comprehensive benchmarks using multiple state-of-the-art YOLO detectors.

**Strengths:**

1. This paper is well-organized and easy to understand.

2. The proposed PIC method offers a training-free, computationally cheap, and effective way to convert simple point annotations into high-fidelity bounding boxes, dramatically reducing annotation cost.

3. The paper provides a very thorough evaluation, benchmarking a wide array of YOLO models for object detection and instance segmentation, establishing strong baselines for future research on this dataset.

**Weaknesses:**

（1）The core innovation is limited, as the work primarily repurposes an existing dataset and relies heavily on existing models like SAM2 for mask generation, while the resulting dataset's annotation quality and reliability remain unvalidated against human annotations.
（2）The proposed PIC method requires manually set hyperparameters, and the lack of a sensitivity analysis or an adaptive mechanism raises concerns about its robustness and generalizability to other domains.
（3）A critical limitation is the absence of comparative experiments against recent point-supervised or weakly-supervised detection methods, making it difficult to assess the true advantage of the PIC approach.
（4）The UAVDB dataset is constructed from a single source, which potentially limits the diversity of scenarios and UAV types compared to incorporating multiple public datasets.
（5）The segmentation pipeline depends entirely on SAM2, whose known limitations on very small objects could introduce systematic errors into the instance masks without proper validation or correction.
（6）The experimental evaluation focuses solely on static image tasks, failing to leverage or demonstrate the dataset's potential for video-based temporal modeling or tracking applications.

**Questions:**

Please see the weaknesses.

---

### Author Response · Authors · 2025-11-19
**Manuscript Withdrawal Notice**

Thank you for the time, effort, and constructive feedback.

After carefully reflecting on the comments, I recognize that several aspects of the current submission can indeed be further strengthened. These include the dependence on SAM2 without sufficient validation, the absence of comparisons with recent point- or weakly-supervised methods, and the heavy reliance on an existing dataset with limited novelty.

Given these limitations, I acknowledge that the work does not yet meet the standard expected for this venue. Therefore, I have decided to withdraw the paper and will continue improving the methodology, evaluation, and dataset construction before preparing a future resubmission.

Thank you again for you review.

---

### Note · Authors · 2025-11-19

**Comment:**

Thank you for the constructive reviews. I am withdrawing the paper at this time and will revise the methodology and experiments accordingly for a future submission.

**Withdrawal Confirmation:**

I have read and agree with the venue's withdrawal policy on behalf of myself and my co-authors.